# THE INFORMATION GEOMETRY OF UNSUPERVISED REINFORCEMENT LEARNING

**Benjamin Eysenbach**[1][2]    **Ruslan Salakhutdinov**[1]    **Sergey Levine**[2][3]

[1]Carnegie Mellon University,    [2]Google Brain,    [3]UC Berkeley
`beysenba@cs.cmu.edu`

## ABSTRACT

How can a reinforcement learning (RL) agent prepare to solve downstream tasks if those tasks are not known a priori? One approach is unsupervised skill discovery, a class of algorithms that learn a set of policies without access to a reward function. Such algorithms bear a close resemblance to representation learning algorithms (e.g., contrastive learning) in supervised learning, in that both are pretraining algorithms that maximize some approximation to a mutual information objective. While prior work has shown that the set of skills learned by such methods can accelerate downstream RL tasks, prior work offers little analysis into whether these skill learning algorithms are optimal, or even what notion of optimality would be appropriate to apply to them. In this work, we show that unsupervised skill discovery algorithms based on mutual information maximization do not learn skills that are optimal for every possible reward function. However, we show that the distribution over skills provides an optimal initialization minimizing regret against adversarially-chosen reward functions, assuming a certain type of adaptation procedure. Our analysis also provides a geometric perspective on these skill learning methods.

## 1 INTRODUCTION

The high sample complexity of reinforcement learning (RL) algorithms has prompted a large body of prior work to study pretraining of RL agents. During the pretraining stage, the agent collects *unsupervised* experience from the environment that is not labeled with any rewards. Prior methods have used this pretraining stage to learn representations of the environment that might assist the learning of downstream tasks. For example, some methods learn representations of the observations (Laskin et al., 2020; Schwarzer et al., 2021) or representations of the dynamics model (Ebert et al., 2018; Ha & Schmidhuber, 2018; Sekar et al., 2020). In this work, we focus on methods that learn a set of potentially-useful policies, often known as *skills* (Salge et al., 2014; Mohamed & Rezende, 2015; Gregor et al., 2016; Achiam et al., 2018; Eysenbach et al., 2018). That is, the learned representation corresponds to a reparametrization of policies. The aim of this unsupervised pretraining is to learn skills that, when a reward function is given, can quickly be combined or composed to maximize this reward. Prior work has demonstrated that this general approach does accelerate learning downstream RL (Gregor et al., 2016; Eysenbach et al., 2018; Achiam et al., 2018). However, prior work offers little analysis about when and where such methods are provably effective. Even simple questions, such as what it means for a set of skills to be optimal, remain unanswered.

Algorithms for unsupervised skill learning are conceptually related to the representation learning methods used to improve supervised learning (Gutmann & Hyvärinen, 2010; Belghazi et al., 2018; Wu et al., 2018; Oord et al., 2018; Hjelm et al., 2018; He et al., 2020). Both typically maximize a lower bound on mutual information, and the learned representations are often combined linearly to solve downstream tasks (Hjelm et al., 2018; Oord et al., 2018). However, whereas prior work in supervised learning has provided thorough analysis of when and where these representation learning methods produce useful features (Kraskov et al., 2004; Song & Ermon, 2019; McAllester & Stratos, 2020), there has been comparatively little analysis into when unsupervised skill learning methods produce skills that are useful for solving downstream RL tasks.

In this paper, we analyze when and where existing skill learning methods based on mutual information maximization are (or are not) optimal for preparing to solve unknown, downstream tasks. On the one hand, we show that the skills learned by these methods are not complete, in that they cannot be used to represent the solution to every RL problem. This result implies that using the learned skills for hierarchical RL may result in suboptimal performance, and suggests new opportunities for better skill learning algorithms. One the other hand, we show that existing methods acquire a policy initialization that is optimal for learning downstream tasks, if that adaptation is performed using an idealized adaptation procedure. To the best of our knowledge, this is the first result showing that unsupervised skill learning methods are optimal in any sense.

Our analysis also illuminates a number of properties of these methods. For example, we show that every skill is optimal for some reward function, and we provide a nontrivial upper bound on the number of unique skills learned. This result implies that these methods cannot learn an infinite number of unique skills, and instead will learn duplicate copies of some skills. The key to our analysis is to view RL algorithms and skill learning algorithms as geometric operations (see Fig. 1). Points correspond to distributions over states, and the set of all possible distributions is a convex polytope that lies on a probability simplex. We show that all reward-maximizing policies lie at vertices of this polytope and that maximizing mutual information corresponds to solving a facility assignment problem on the simplex.

The main contribution of this paper is a proof that skill learning algorithms based on mutual information are optimal for minimizing regret against unknown reward functions, assuming that adaptation is performed using a certain procedure. Our proof of optimality relies on certain problem assumptions, leaving the door open for future skill learning algorithms to perform better under different problem assumptions. This contribution provides a rigorous notion of what it means for an unsupervised RL algorithm to be optimal, and also answers additional questions about unsupervised skill learning algorithms, such as whether the skills correspond to reward-maximizing policies and how many unique skills will be learned.

## 2 PRIOR WORK

In the *unsupervised RL* setting, an agent interacts with the environment without access to a reward function, with the aim of learning some representation of the environment that will assist in learning downstream tasks. Prior work has proposed many approaches for this problem, including learning a dynamics model (Ebert et al., 2018; Ha & Schmidhuber, 2018; Sekar et al., 2020), learning compact representations of observations (Laskin et al., 2020; Schwarzer et al., 2021), performing goal-conditioned RL without hand-specified rewards (Eysenbach et al., 2020; Chebotar et al., 2021; Schwarzer et al., 2021), doing pure exploration (Salge et al., 2014; Mohamed & Rezende, 2015; Pathak et al., 2017; Lee et al., 2019b; Hazan et al., 2019; Seo et al., 2021) or learning collections of skills (Gregor et al., 2016; Eysenbach et al., 2018; Achiam et al., 2018; Warde-Farley et al., 2018; Sharma et al., 2019; Hansen et al., 2020; Campos et al., 2020). These prior methods are similar in that they all learn *representations*, with different methods learning representation of observations, dynamics, or policies. While each approach works well in some setting (see Laskin et al. (2021) for a recent benchmark), the precise connection between these pretraining methods and success on downstream tasks remains unclear. We will focus on unsupervised skill learning methods.

The problem of unsupervised pretraining for RL has also been studied in the RL theory community (Agarwal et al., 2020; Misra et al., 2020; Modi et al., 2021), where it is referred to as the *reward-free* setting. The algorithms proposed in these prior works use a version of goal-conditioned RL for pretraining. However, the aim of these methods is different from ours: these methods aim to collect a sufficient breadth of data, whereas we focus on learning good representations.

Our paper is similar to prior work that uses a geometric perspective to analyze RL algorithms (Dadashi et al., 2019; Bellemare et al., 2019). For example, Dadashi et al. (2019) visualize the value function as a point in a high-dimensional space, where each coordinate indicates the value of one state. In contrast, our analysis will visualize policies as points, where each coordinate indicates the probability of visiting that state. This difference allows us to analyze the unsupervised RL setting, where we cannot define a value function because the reward function is unknown. Our analysis uses ideas from the field of information geometry (Amari & Nagaoka, 2000). Prior work has parametrized the RL problem in terms of the state distribution (Puterman, 1990, Eq. 6.9.2), and proposed RL algorithms that solve for the optimal state distribution (Ng et al., 1999; Wang et al., 2007; Nachum & Dai, 2020).

Our analysis uses a similar procedure for adapting to new reward functions after the unsupervised learning stage.

## 3 PRELIMINARIES

We focus on infinite-horizon MDPs with discrete states $\mathcal{S}$ and actions $\mathcal{A}$, initial state distribution $p_0(\mathbf{s_0})$, dynamics $p(\mathbf{s_{t+1}} \mid \mathbf{s_t}, \mathbf{a_t})$, and discount factor $\gamma \in (0, 1)$. We assume a Markovian policy $\pi(\mathbf{a} \mid \mathbf{s})$, and define the discounted state occupancy measure of this policy as $\rho^\pi(\mathbf{s}) = (1 - \gamma) \sum_{t=0}^\infty \gamma^t P_t^\pi(\mathbf{s})$, where $P_t^\pi(\mathbf{s})$ is the probability that policy $\pi$ visits state $\mathbf{s}$ at time $t$. We define $\mathcal{C}$ to be the set of state marginal distributions that are feasible under the environment dynamics. The RL objective can be expressed using the reward function $r(\mathbf{s})$ and the state marginal distribution:

$$(1 - \gamma)\mathbb{E}_\pi \left[ \sum_{t=0}^\infty \gamma^t r(\mathbf{s_t}) \right] = \mathbb{E}_{\rho^\pi(\mathbf{s})} \left[ r(\mathbf{s}) \right].$$

The factor of $1 - \gamma$ accounts for the fact that the sum of discount factors $1 + \gamma + \gamma^2 + \cdots = \frac{1}{1-\gamma}$. Without loss of generality, we focus on *state-dependent reward functions*; action-dependent reward functions can be handled by modifying the state to include the previous action. While our analysis will ignore function approximation error, it applies to any MDP, including MDPs where observations correspond to features (e.g., activations of a frozen neural network) rather than original states.

### 3.1 UNSUPERVISED SKILL LEARNING ALGORITHMS

Prior skill learning algorithms learn a policy $\pi_\theta(\mathbf{a} \mid \mathbf{s}, \mathbf{z}_{\text{input}})$ with parameters $\theta$ and conditioned on an additional input $\mathbf{z}_{\text{input}} \sim p(\mathbf{z}_{\text{input}})$. Let $\rho^{\pi_\theta}(\mathbf{s} \mid \mathbf{z}_{\text{input}})$ denote the state marginal distribution of policy $\pi_\theta(\mathbf{a} \mid \mathbf{s}, \mathbf{z}_{\text{input}})$. We will focus on methods (Florensa et al., 2017; Eysenbach et al., 2018) that maximize the mutual information between the representation $\mathbf{z}_{\text{input}}$ and the states $\mathbf{s}$ visited by policy $\pi_\theta(\mathbf{a} \mid \mathbf{s}, \mathbf{z}_{\text{input}})$:

$$\max_{\theta, p(\mathbf{z}_{\text{input}})} I(\mathbf{s}; \mathbf{z}_{\text{input}}) \triangleq \mathbb{E}_{p(\mathbf{z}_{\text{input}})} \mathbb{E}_{\rho^{\pi_\theta}(\mathbf{s}|\mathbf{z}_{\text{input}})} [\log \rho^{\pi_\theta}(\mathbf{s} \mid \mathbf{z}_{\text{input}}) - \log \rho^{\pi_\theta}(\mathbf{s})]. \tag{1}$$

We will refer to such methods as mutual information skill learning (MISL), noting that our analysis might be extended to other mutual information objectives (Salge et al., 2014; Mohamed & Rezende, 2015; Achiam et al., 2018; Sharma et al., 2019). Because these methods have two learnable components, $\theta$ and $p(\mathbf{z}_{\text{input}})$, analyzing them directly can be challenging. Instead, we will use a simplified abstract model of skill learning where both the policy parameters $\theta$ and the latent variable $\mathbf{z}_{\text{input}}$ are part of a single representation, $\mathbf{z} = (\theta, \mathbf{z}_\theta)$. Then, we can define a skill learning algorithm as learning a single distribution $p(\mathbf{z})$:

$$\max_{p(\mathbf{z})} I(\mathbf{s}; \mathbf{z}) \triangleq \mathbb{E}_{p(\mathbf{z})} \mathbb{E}_{\rho(\mathbf{s}|\mathbf{z})} [\log \rho(\mathbf{s} \mid \mathbf{z}) - \log \rho(\mathbf{s})]. \tag{2}$$

We can recover Eq. 1 by choosing a distribution $(\mathbf{z})$ that factors as $p(\mathbf{z} = (\theta, \mathbf{z}_{\text{input}})) = \delta(\theta)p(\mathbf{z}_{\text{input}})$. Whereas skills learning algorithms are often viewed as optimizing individual skills, this simplified abstract model lets us think about these algorithms as assigning different probabilities to different policies a probability of zero to almost every skill. We will refer to the small number of policies that are given non-zero probability as the "skills." We will show that, in general, the learned skills fail to cover all possible optimal policies. However, we show that the *distribution* over skills that maximizes mutual information, when converted into a distribution over states, provides the best state distribution for optimizing an adversarially-chosen reward function using an idealized optimizer. In general, this idealized optimizer is infeasible to implement, suggesting that existing skill learning methods do not provide the best initialization for practical optimization methods.

## 4 OVERVIEW OF MAIN RESULTS

Our main results analyze the relationship between the skills learned via the mutual information objective in Equation 2 and optimal policies. We will show that, in the general case, maximizing mutual information alone does not result in skills that cover the set of all optimal policies, *no matter*

*how many skills are learned.* Perhaps surprisingly, when the number of skills is large enough, additional skills no longer differentiate from the existing ones, despite some potentially optimal policies not being covered. However, we will show that the the distribution over skills learned by MISL is optimal in a different sense: when this distribution over skills is converted into a distribution over states, it provides a prior over states that is close to the state distributions of reward-maximizing policies. This prior is optimal in the sense that, when combined with a particular adaptation procedure, it minimizes regret against the worst-case reward function. We formally state this result below:

**Theorem 4.1.** *The optimal initialization for adapting to an unknown reward function is the average state marginal given by maximizing mutual information (Eq. 2):*

$$\min_{\rho(\mathbf{s}) \in \mathcal{C}} \max_{r(\mathbf{s}) \in \mathbb{R}^{|\mathcal{S}|}} \text{ADAPTATIONOBJECTIVE}(\rho(\mathbf{s}), r(\mathbf{s})) = \max_{p(\mathbf{z})} I(\mathbf{s}; \mathbf{z}),$$

*where the adaptation objective is defined as*

$$\text{ADAPTATIONOBJECTIVE}(\rho(\mathbf{s}), r(\mathbf{s})) \triangleq \min_{\rho^*(\mathbf{s}) \in \mathcal{C}} \underbrace{\max_{\rho^+(\mathbf{s}) \in \mathcal{C}} \mathbb{E}_{\rho^+(\mathbf{s})}[r(\mathbf{s})] - \mathbb{E}_{\rho^*(\mathbf{s})}[r(\mathbf{s})]}_{regret} + D_{\text{KL}}(\rho^*(\mathbf{s}) \parallel \rho(\mathbf{s})).$$

(3)

*Moreover, the optimal prior can be written as the average state distribution of the skills:* $\rho^*(\mathbf{s}) = \mathbb{E}_{p^*(\mathbf{z})}[\rho(\mathbf{s} \mid \mathbf{z})].$

Note that the optimal prior, $\rho^*(\mathbf{s})$ is not hard to compute: it corresponds to sampling one of the skills $\mathbf{z} \sim p^*(\mathbf{z})$ and then execute the corresponding policy, $\pi(\mathbf{a} \mid \mathbf{s}, \mathbf{z})$.

The regularized regret objective, ADAPTATIONOBJECTIVE, can be interpreted in several different ways. The regularization can be motivated as preventing overfitting to a limited number of samples of a noisy reward function. We could also interpret this term as reflecting the cost of adapting from to the new task, but under a somewhat idealized model of the learning process. Precisely, the adaptation objective is equivalent to performing one step of natural gradient descent (Amari, 1998; Kakade, 2001; Martens,

Table 1: Notation for analysis.

| | |
|---|---|
| $r(\mathbf{s})$ | Reward function. |
| $\rho(\mathbf{s})$ | Initialization. |
| $\rho^+(\mathbf{s})$ | Optimal state marginal distribution for reward $r(\mathbf{s})$. Used to define regret. |
| $\rho^*(\mathbf{s})$ | Optimal state marginal distribution after adaptation, which maximizes reward and minimizes information cost. |

2020), where the underlying metric is the state marginal distribution (Raskutti & Mukherjee, 2015, Theorem 1). This natural gradient is different from the standard natural policy gradient (Kakade, 2001), which does not depend on the environment dynamics and is much easier to estimate. This equivalence suggests that our notion of adaptation is idealized: it assumes that this adaptation step can be performed exactly, without considering how data is collected, how state marginal distributions are estimated, or how the underlying policy parameters should be updated. Nonetheless, this paper is (to the best of our knowledge) the first to provide any formal connection between the mutual information objective optimized by unsupervised skill learning algorithms and performance on downstream tasks.

## 5 RL AS GEOMETRY ON A SIMPLEX

In order to prove our main result and further analyze the kinds of policies learned by unsupervised skill learning methods, we will first introduce a geometric perspective on RL. This perspective will allow us to better understand how unsupervised skill learning relates to the space of potentially optimal policies (for state-dependent reward functions).

We visualize policies by their discounted state occupancy measure, a $|\mathcal{S}|$-dimensional point lying on the probability simplex (see Fig. 1). Note that this is a distribution over states, not actions, so a policy that chooses actions uniformly will not necessarily lie at the center of the simplex. Of course, not all state marginals are feasible, as the MDP dynamics might preclude certain distributions. We will be interested in where skill learning algorithms place skills on this probability simplex, and whether these skills are optimal for learning downstream tasks. Note that not all points on the simplex correspond to realizable policies, and multiple policies could map to the same state occupancy measure.

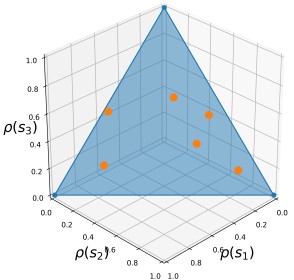 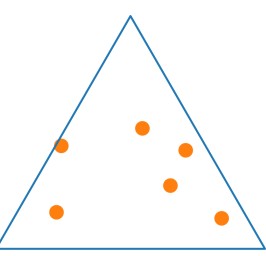 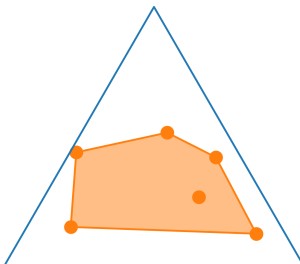

Figure 1: **Polytope of achievable policies**: *(Left)* For a 3-state MDP, the set of feasible state marginal distributions is described by a triangle $[(1, 0, 0), (0, 1, 0), (0, 0, 1)]$ in $\mathbb{R}^3$. We plot the state occupancy measure of 6 different policies. *(Center)* We plot those same policies, but remove the coordinate axes to improve visual clarity. *(Right)* For any convex combination of valid state occupancy measures, there exists a Markovian policy that has this state occupancy measure. Thus, policy search happens inside a convex polytope.

## 5.1 WHICH STATE MARGINALS ARE ACHIEVABLE?

Understanding which state marginals are achievable is important for analyzing what unsupervised skill learning algorithms do, and how the learned skills relate to downstream *state-dependent* reward functions. While the set of achievable state marginal distributions can be described by a set of linear equations (Puterman, 1990, Eq. 6.9.2) (see Appendix A.1), we offer a different and complementary description based on the following property (Ziebart, 2010, Theorem 2.8):

**Proposition 1.** *Given a set of feasible state marginals, any convex combination is also feasible.*

This property is illustrated in Fig. 1 (right): if we plot the state marginals for six policies, then any point inside the convex hull of those state marginals is also feasible. That is, there exists a Markovian policy that achieves that state marginal distribution. In general, we can obtain the set of *all* feasible state marginals by taking the convex hull of the state marginals of all deterministic policies. We will refer to this set of achievable state marginals as a *polytope*. Every vertex of the state marginal polytope will *contain* a deterministic policy,[1] and the state marginal of every policy can be represented as a convex combination of the state marginals at the vertices.

## 5.2 WHICH POLICIES ARE OPTIMAL?

We now analyze where reward-maximizing policies lie within this state marginal polytope. Again taking a geometric perspective, we visualize reward functions as vectors starting from the origin, so that the expected return objective can be expressed as the inner product between the state marginal distribution and the reward vector (see Fig. 2). This visualization helps illustrate two facts about the state marginal distributions of reward-maximizing policies:

**Proposition 2.** *For every state-dependent reward function, among the set of policies that maximize that reward function is one that lies at a vertex of the state marginal polytope.*

**Proposition 3.** *For every vertex of the state marginal polytope, there exists a reward function for which that vertex corresponds to a reward-maximizing policy.*

Proposition 2 is an application of the maximum principle (Rockafellar, 2015, Chpt. 32); Proposition 3 is an application of the strong separation theorem (Hiriart-Urruty & Lemaréchal, 2004, Theorem 3.2.2). These observations

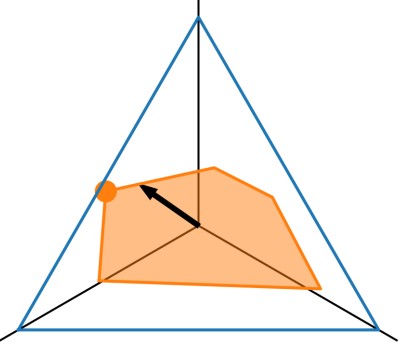

Figure 2: **Maximizing Rewards:** We visualize reward functions as vectors starting at the origin. Maximizing expected return corresponds to maximizing the dot product between the state marginal distribution and this reward vector.

suggest that, for the purpose of RL, it is

---

[1]Note, however, that some deterministic policies may not lie at vertices of this polytope.

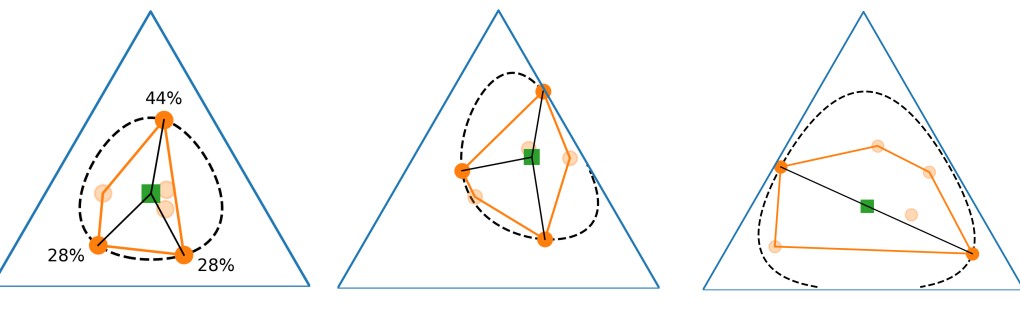

(a) Examples where MISL recovers $3 = |\mathcal{S}|$ skills.

(b) Example where MISL learns only two unique skills.

Figure 3: **Information Geometry of Skill Learning**: Maximizing mutual information is equivalent to minimizing the *maximum* divergence between a prior distribution over states $p(\mathbf{s})$ (green square) and any achievable state marginal distribution (Lemma 6.5). MISL learns a distribution over state distributions, $p(\mathbf{z})$, that assigns non-zero probability to only a small number of state distributions (solid orange circles).

sufficient to search among policies whose state marginals lie at vertices.[2] These observations also suggests an objective for unsupervised skill learning:

**Definition 5.1** (Vertex discovery problem). Given a controlled Markov process (i.e., an MDP without a reward function), find the smallest set of policies such that every vertex of the state marginal polytope contains at least one policy.

While we are unaware of any tractable algorithm for exactly solving this problem, the next section discusses how a prior skill learning algorithm provides an efficient *approximation* to this problem.

## 6 THE GEOMETRY OF SKILL LEARNING

Now that we know where reward-maximizing policies lie on the polytope, we now analyze where the state marginal distributions of skills lie on the polytope. Our main result is that mutual information skill learning recovers some, but not all, of the reward-maximizing policies. We show that the skills do not cover every vertex, and are thus insufficient for representing the solution to every downstream RL task. We then will show that the *average* state distribution of the skills provides an optimal initialization for solving downstream tasks using a certain adaptation procedure.

The analysis in this section determines where skills lie on the polytope by analyzing the KL divergence between points on the polytope. This analysis does not assume that the KL divergence is symmetric (which it is not) or that it obeys the triangle inequality (which it does not).

### 6.1 WHERE DO SKILLS LEARNED BY MUTUAL INFORMATION FALL ON THE POLYTOPE?

We analyze where the skills learned by MISL lie on the state marginal polytope by dissecting the mutual information objective. The mutual information can be written as the average divergence between each policy's state distribution $\rho(\mathbf{s} \mid \mathbf{z})$ and the average state distribution, $\rho(\mathbf{s})$:

$$I(\mathbf{s}; \mathbf{z}) = \mathbb{E}_{p(\mathbf{z})}\left[D_{\mathrm{KL}}(\rho(\mathbf{s} \mid \mathbf{z}) \parallel \rho(\mathbf{s}))\right].$$

To optimize this mutual information, a skill learning algorithm would choose to assign higher probability to some policies and lower probability (perhaps zero) to other policies. In particular, this mutual information suggests that skill learning algorithms should assign higher probabilities to policies whose state distributions are further from the average state distribution. In fact, if one policy has a state distribution that is further to the average state distribution than another policy, then the skill learning algorithm should assign a probability of zero to this second policy. Thus, the net result of optimizing mutual information is to place probability only on policies that are as far away from their average as possible:

---

[2]While it is well known that any reward function has a deterministic optimal policy (Puterman, 1990, Sec. 4.4), our visualization adds nuance to the story, suggesting that searching in the space of *all* deterministic policies can be wasteful, at least in the case of rewards that depend only on state.

**Lemma 6.1.** *The MISL skills must lie at vertices of the state marginal polytope.*

This result, which we confirm experimentally[3] in Fig. 3, answers the question of where the skills learned by mutual information fall on the polytope. Recalling that the vertices of the state marginal polytope correspond to optimal policies (Proposition 2), this result implies that the skills learned by MISL are a subset of the set of optimal policies. Thus,

**Corollary 6.1.1.** *MISL skills are all optimal for some downstream state-dependent reward functions.*

## 6.2 How Many Unique Skills are Learned?

A simple way to use skills acquired during pretraining to solve a downstream RL task is to simply search among the skills to find the one that achieves the highest reward. So, it is natural to ask whether skill learning algorithms learn skills that are optimal for every possible reward function. Indeed, this is a natural notion of what it might mean for a skill learning algorithm to be optimal. Prior work motivates the use of larger dimensional $\mathbf{z}$ or continuous-valued $\mathbf{z}$, arguing that these design decisions will allow the method to learn a larger (perhaps infinite) number of skills (Eysenbach et al., 2018; Hausman et al., 2018). However, it has remained unclear whether such design decisions are useful and whether the goal of acquiring an infinite or exponential number of skills is even possible. Among prior skill learning methods, we are unaware of prior work that characterizes the *number* of skills that these methods will learn. In this section we show that existing skill learning algorithms based on mutual information are not optimal in this sense. Our proof will follow a simple counting argument based on the number of unique skills.

To count the number of unique skills, we start by analyzing the distance between each skill and the state distribution averaged over skills, measured as a KL divergence between state marginals. We can use the KKT conditions to prove that all skills with non-zero support (i.e., for which $p(\mathbf{z}) > 0$) have an equal divergence from the average state marginal:

**Lemma 6.2.** *Let $\rho(\mathbf{s} \mid \mathbf{z})$ be given (not learned). Let $p(\mathbf{z})$ be the (learned) solution to maximizing mutual information (Eq. 2) and let $\rho(\mathbf{s}) = \mathbb{E}_{p(\mathbf{z})}[\rho(\mathbf{s} \mid z)]$ be the average state marginal. Then the following holds:*

$$p(\mathbf{z}) > 0 \implies D_{\mathrm{KL}}(\rho(\mathbf{s} \mid \mathbf{z}) \| \rho(\mathbf{s})) = \max_{\mathbf{z}^*} D_{\mathrm{KL}}(\rho(\mathbf{s} \mid \mathbf{z}^*) \| \rho(\mathbf{s})) \triangleq d_{max}.$$

We visualize this result in Fig. 3: the dashed line denotes a state marginals $\rho(\mathbf{s} \mid \mathbf{z})$ that are equidistant from the average state marginal (green square). Note that all skills where MISL places non-zero probability mass (denoted by solid orange dots instead of transparent orange dots) lie on this dashed circle.

This lemma allows us to count how many unique skills MISL acquires. To state the result formally, we need to assume that the vertices of the state marginal polytope do not end up in a special concyclic arrangement. This assumption, which we formally define in Appendix A.7, is similar to the collinearity assumption in linear regression (Mitchell & Beauchamp, 1988) and occurs with measure zero.

**Lemma 6.3.** *Under Assumption 1, MISL will recover at most $|\mathcal{S}|$ distinct skills.*

*Proof.* The proof follows a simple counting argument. Without loss of generality, we only consider vertices of the state marginal polytope as candidate skills (Lemma 6.1). The location of the average state marginal ($\rho(\mathbf{s})$) on the probability simplex is determined by the location of the skills and the distance to the skills. The average state marginal is a vector in $|\mathcal{S}|$-dimensional space with $|\mathcal{S}| - 1$ degrees of freedom; we subtract one to account for the constraint that the state marginal sum to 1.

Consider adding skills one by one. When we add a new skill (i.e., set $p(\mathbf{z}) > 0$ for some $\mathbf{z}$), we add an additional constraint to the location of this average state marginal; namely, that $D_{\mathrm{KL}}(\rho(\mathbf{s} \mid \mathbf{z}) \| p(\mathbf{z})) = d_{\max}$. The value of $d_{\max}$ adds one additional degree of freedom. Thus, after adding $(|\mathcal{S}| - 1) + 1$ skills, the average state marginal is fully specified. Adding additional skills would make the average state marginal ill-defined, as it would have to violate one of the distance constraints. Thus, we can have at most $|\mathcal{S}|$ unique skills. $\qquad\square$

---

[3]Code to reproduce: `https://github.com/ben-eysenbach/info_geometry/blob/main/experiments.ipynb`

We verify this result with a simple experiment. We randomly generate tabular MDPs and learn 100 (possibly redundant) skills using MISL. As shown in Fig. 4, the number of *unique* skills for an MDP is never greater than the number of states in that MDP, supporting Lemma 6.3.

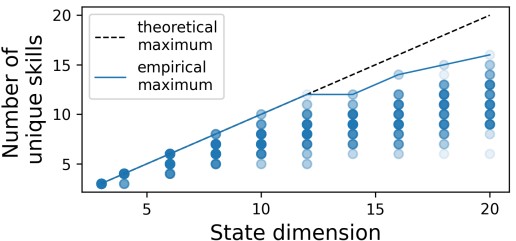

MISL need not learn exactly $|\mathcal{S}|$ skills. In some MDPs, such as the one illustrated in Fig. 3b, MISL acquires only 2 unique skills for a 3-state MDP. If an additional skill (at a vertex) were added, then the average distance from the skills to the average state marginal would decrease,

Figure 4: As predicted by our theory, the number of unique skills learned by MISL is upper bounded by the number of states.

worsening the mutual information objective. Additionally, when the regularity assumption is violated, there could be more than $|\mathcal{S}|$ skills that all have a KL divergence of $d_{\max}$ from the initialization $\rho(s)$, in which case MISL could learn more than $|\mathcal{S}|$ skills.

If MISL can only learn up to $|\mathcal{S}|$ unique skills, will it learn a skill for maximizing every possible reward function? While some environments have exactly $|\mathcal{S}|$ optimal policies, general environments can have up to $|\mathcal{A}|^{|\mathcal{S}|}$ optimal policies (see Appendix A.8). Thus, MISL does not solve the vertex discovery problem:

**Theorem 6.4.** *In general, skill learning algorithms that maximize the mutual information $I(\mathbf{s}; \mathbf{z})$ will not recover the set of all optimal policies.*

This result is important for determining how to use the output of an unsupervised skill learning algorithm to solve downstream tasks. One approach to adaptation is to find whichever skill receives the highest return. This result indicates that this approach will produce suboptimal performance on a large number of reward functions. In fact, for every reward maximizing policy which a skill learning algorithm can represent, there can be an exponential number that the skill learning algorithm can fail to represent. This result casts doubt on the use of the skills themselves for solving downstream tasks, at least those skills learned by the existing generation of skill learning algorithms.

Might other simple algorithms for skill discovery recover the entire set of optimal policies? We are currently unaware of any algorithm that does solve the vertex discovery problem. In Appendix A.5, we show that *(1)* including actions in the mutual information objective or *(2)* maximizing a collection of indicator reward functions also fail to discover all optimal policies.

Unsupervised skill learning algorithms learn a full distribution over skills. Instead of just looking at the set of policies that are given non-zero probability (i.e., skills), do the numerical values of these probabilities give us useful information to help solve downstream tasks? In the next section, we discuss how our main result (Theorem 4.1) helps to address this equestion.

### 6.3 An Optimal Initialization for an Unknown Reward Function

We now discuss a different perspective on optimizing mutual information, thinking about the *average* state distribution learned by MISL and whether it is optimal for solving downstream tasks. This perspective provides a proof sketch of our main result, Theorem 4.1.

Intuitively, the difficulty of solving downstream tasks depends on how much the agent must adapt its initialization to learn the optimal policy for downstream tasks. Different initializations will make some tasks easier and some tasks more difficult. If we think about adaptation as happening in the space of state marginal distributions, then there is a precise relationship between maximizing mutual information and minimizing the amoung of adaptation required.

**Lemma 6.5** (Theorem 13.11 from Cover & Thomas (2006), based on Gallager (1979); Ryabko (1979)). *Let $\rho(\mathbf{s} \mid \mathbf{z})$ be given. Maximizing mutual information is equivalent to minimizing the divergence between the average state distribution $\rho(\mathbf{s})$ and the state distribution of the furthest skill:*

$$\max_{p(\mathbf{z})} I(\mathbf{s}; \mathbf{z}) = \min_{\rho(\mathbf{s})} \max_{\mathbf{z}} D_{\mathrm{KL}}(\rho(\mathbf{s} \mid \mathbf{z}) \parallel \rho(\mathbf{s}))$$

This characterization can be understood as a *facility location problem* (Farahani & Hekmatfar, 2009) or a *1-center problem* (Minieka, 1970; Garfinkel et al., 1977). Since maximizing mutual information

learns an average state distribution that is close to other policies, we expect that this average state distribution will be a good initialization for learning these other policies. This result provides the intuition behind the proof of Theorem 4.1 in Appendix A.6.

**Relationship with meta-learning.** The pretraining objective (Eq. 3) bears a resemblance to the objective for meta-learning algorithms (Franceschi et al., 2018; Lee et al., 2019a; Rajeswaran et al., 2019): both aim to find a parameter initialization for which an inner optimization problem can be efficiently solved. Whereas these methods assume access to a training distribution, skill learning algorithms used for pretraining do not have information about the downstream task. Moreover, meta-learning typically studies the average performance at test time, our objective measures the worst-case performance at test time. From this perspective, unsupervised skill learning could be interpreted as optimal pretraining for an idealized downstream adaptation procedure that searches over policies in the space of state marginals. Of course, this is an abstraction of practical RL methods, which must search in the space of policy parameters. This connection suggests that it might be possible to design meta-learning algorithms using the tools of mutual information skill learning, but the underlying mutual information objective might need to be modified so that adaptation can happen in the space of policy parameters, rather than state marginal distributions.

## 6.4 IMPLICATIONS FOR FUTURE SKILL LEARNING ALGORITHMS

Our analysis sheds light on fundamental limitations of skill learning algorithms based on mutual information. First, as explained in Sec. 6.2, existing algorithms based on mutual information do not learn skills that are sufficient for representing every potentially optimal policy. Future skill learning algorithms that can represent every policy may prove better at adapting to new tasks.

Our analysis shows that MISL produces an optimal initialization for a particular adaptation procedure: natural gradient descent in the space of state marginal distribution. This natural gradient is not the standard natural policy gradient (Kakade, 2001), which is performed in policy parameter space, but rather the natural gradient with respect to the state marginal distribution. This is a very strong (hypothetical) reinforcement learning algorithm, because it explores directly in the space of state marginals. This gradient implicitly depends on the environment dynamics, and computing this gradient would likely require exploring the environment to determine how changes in a policy's parameters affect changes in the states visited. Said in other words, our analysis suggests that existing unsupervised skill learning methods are optimal for a somewhat unrealistic and idealized model of how downstream policy learning is performed. This suggests an opportunity to develop significantly more powerful algorithms that serve as effective initializations for more realistic adaptation procedures, such as those studied in meta-RL (Wang et al., 2016; Finn et al., 2017; Humplik et al., 2019).

## 6.5 LIMITATIONS OF ANALYSIS

In our analysis, we used a simplified model of skill learning algorithms, one that viewed policy parameters and latent codes as one monolithic representation, with different skills using entirely different sets of parameters. Practical implementations of skill learning algorithms are less expressive, which may change the geometry of the state marginal polytope. For example, a limited policy class may be insufficient to represent some state marginal distributions, introducing "holes" in the state marginal polytope, or splitting the polytope into two disjoint regions. It remains to be seen whether these practical implementations actually learn fewer unique skills, and how these expressivity constraints affect the average state distribution.

## 7 CONCLUSION

We have used a geometric perspective to answer a number of open questions in the study of skill discovery algorithms and unsupervised RL. We also showed that the skills learned by unsupervised skill learning algorithms are not sufficient for maximizing every downstream reward function. However, we do show that the average state distribution of the skills provides an initialization that is optimal for solving unknown downstream tasks using a particular adaptation procedure. Our analysis showed that the number of skills is bounded by the number of states, suggesting that scaling these methods to learn more skills will hit a theoretical upper bound. Taken together, we believe that this analysis sheds light on what prior methods are doing, when and where they should work, and what limitations could be addressed in the next generation of skill learning algorithms.

**Acknowledgements.** We thank Abhishek Gupta, Alex Alemi, Archit Sharma, Jonathan Ho, Michael Janner, Ruosong Wang, Shane Gu and anonymous reviewers for discussions and feedback on the paper. This material is supported by the Fannie and John Hertz Foundation and the NSF GRFP (DGE1745016).

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

## A    PROOFS AND ADDITIONAL ANALYSIS

### A.1    MATHEMATICAL DEFINITION OF THE STATE MARGINAL POLYTOPE

We provide a formal description of the constraint set. Precisely, the set of achievable state marginal distributions are those for which the *state-action* marginal distribution $\rho(\mathbf{s}, \mathbf{a})$ satisfies the following linear constraints (Puterman, 1990, Eq. 6.9.2):

$$\sum_{\mathbf{a}' \in \mathcal{A}} \rho(\mathbf{s}', a') = (1 - \gamma)p_1(\mathbf{s}') + \gamma \sum_{\mathbf{s} \in \mathcal{S}, \mathbf{a} \in \mathcal{A}} p(\mathbf{s}' \mid \mathbf{s}, \mathbf{a})\rho(\mathbf{s}, \mathbf{a}) \quad \forall \mathbf{s}' \in \mathcal{S}$$

$$\rho(\mathbf{s}, \mathbf{a}) \geq 0 \quad \forall \mathbf{s} \in \mathcal{S}, \mathbf{a} \in \mathcal{A}.$$

The constraint that $\rho$ be a valid probability distribution (i.e., $\sum_{\mathbf{s}, \mathbf{a}} \rho(\mathbf{s}, \mathbf{a}) = 1$) is redundant with the constraints above. These constraints can also be written in matrix form as $(\tilde{I} - \gamma P)\vec{\rho}_{s,a} = (1 - \gamma)\vec{p}_1$, where $\tilde{I}$ is a binary matrix that sums the action probabilities for each state (i.e., $\tilde{I}\vec{\rho}_{s,a} = \vec{\rho}_s$), $P$ is the transition matrix, and $\vec{p}_1$ is the initial state distribution.

### A.2    ADDITIONAL INTUITION FOR THE STATE MARGINAL POLYTOPE

This section provides some more intuition for what the state marginal polytope represents by examining a number of properties. First, note that a policy that chooses actions uniformly does not necessarily achieve a uniform state occupancy measure. For example, if actions directly control the next state, then duplicating one action would cause a random policy to shift from the center of the simplex. As a second example, one state may simply not be reachable.

Second, note that two policies can have the same state occupancy measure. This case can happen if (say) multiple actions have similar effects or two agents visit similar states in different orders. Note that, from the perspective of reward maximizing, the state occupancy measure is a sufficient representation of policies (i.e., it is an effect method for state aggregation (Van Roy, 2006; Singh et al., 1995)).

Finally, because some states may be more challenging to reach than others, not all state occupancy measures are realizable: some states may be challenging to reach, meaning that the agent cannot spend more than a small fraction of the episode at that state. A consequence of this fact is that we will restrict our attention to a subset of the state occupancy measures on the probability simplex. Our next section analyzes which state occupancy measures are achievable.

### A.3    CONVEX COMBINATIONS OF FEASIBLE STATE MARGINALS ARE FEASIBLE

This section proves that convex combinations of feasible state marginal distributions are also feasible. First, let two policies $\pi_1(\mathbf{a} \mid \mathbf{s})$ and $\pi_2(\mathbf{a} \mid \mathbf{s})$ be given and consider their corresponding state marginals, $\rho_1(\mathbf{s})$ and $\rho_2(\mathbf{s})$. We illustrate these marginals in Fig. 1 (center). Any state marginal on the line between the two state marginals $\rho_1$ and $\rho_2$ is also achievable. We can express all such state marginals as $\rho_\lambda = \lambda\rho_1 + (1 - \lambda)\rho_2$. One way to construct the policy that achieves the state marginal $\rho_\lambda$ is to form a *non-Markovian* mixture policy: sample $i \sim \text{BERNOULLI}(\lambda)$ at the start of

each episode and then sample actions $\mathbf{a} \sim \pi_i(\mathbf{a} \mid \mathbf{s})$ for each step of that episode. By construction, this mixture policy will have a state occupancy measure of $\rho_\lambda$.

It turns out that there also exists a *Markovian* policy $\pi_\lambda$ that achieves the state marginal $\rho_\lambda$:

$$\pi_\lambda(\mathbf{a} \mid \mathbf{s}) = \lambda(\mathbf{s})\pi_1(\mathbf{a} \mid \mathbf{s}) + (1 - \lambda(\mathbf{s}))\pi_2(\mathbf{a} \mid \mathbf{s}) \quad \text{where} \quad \lambda(\mathbf{s}) \triangleq \frac{\lambda\rho_1(\mathbf{s})}{\lambda\rho_1(\mathbf{s}) + (1 - \lambda)\rho_2(\mathbf{s})} \quad (4)$$

While the existence proof is somewhat involved (Ziebart, 2010, Theorem 2.8)(Feinberg & Shwartz, 2012, Theorem 6.1), the construction of the policy is surprisingly simple. First, note that it is sufficient to ensure $\rho_\lambda(\mathbf{s}, \mathbf{a}) = \lambda\rho_1(\mathbf{s}, \mathbf{a}) + (1 - \lambda)\rho_2(\mathbf{s}, \mathbf{a})$. Then, we apply Bayes' rule to determine the corresponding policy:

$$\begin{aligned}
\pi_\lambda(\mathbf{a} \mid \mathbf{s}) = \frac{\rho_\lambda(\mathbf{s}, \mathbf{a})}{\rho_\lambda(\mathbf{s})} &= \frac{\lambda\rho_1(\mathbf{s}, \mathbf{a}) + (1 - \lambda)\rho_2(\mathbf{s}, \mathbf{a})}{\lambda\rho_1(\mathbf{s}) + (1 - \lambda)\rho_2(\mathbf{s})} \\
&= \frac{\lambda\pi_1(\mathbf{a} \mid \mathbf{s})\rho_1(\mathbf{s}) + (1 - \lambda)\pi_2(\mathbf{a} \mid \mathbf{s})\rho_2(\mathbf{s})}{\lambda\rho_1(\mathbf{s}) + (1 - \lambda)\rho_2(\mathbf{s})} \\
&= \lambda(\mathbf{s})\pi_1(\mathbf{a} \mid \mathbf{s}) + (1 - \lambda(\mathbf{s}))\pi_2(\mathbf{a} \mid \mathbf{s}) \quad \text{where} \quad \lambda(\mathbf{s}) \triangleq \frac{\lambda\rho_1(\mathbf{s})}{\lambda\rho_1(\mathbf{s}) + (1 - \lambda)\rho_2(\mathbf{s})}.
\end{aligned}$$

By induction, one can also show that convex combinations of *multiple* state marginals also yields achievable state marginal distributions. We illustrate this observation in Fig. 1 (right). The construction of the corresponding policy is analogous to Eq. 4.

### A.4 PROOF OF LEMMA 6.1

We now prove Lemma 6.1:

*Proof.* If $p(\mathbf{z})$ were non-zero for a policy where the state marginal $\rho(\mathbf{s} \mid \mathbf{z})$ did not lie at a vertex, then the mutual information objective could be improved by shifting $p(\mathbf{z})$ to place probability mass on the vertex rather than interior point $\rho(\mathbf{s} \mid \mathbf{z})$. $\square$

### A.5 ALTERNATIVE STRATEGIES ALSO FAIL TO DISCOVERY ALL VERTICES

**Indicator reward functions.** Another simple method for skill discovery, based in prior work (Misra et al., 2020, Sec. 3.1), is to define a sparse reward function for visiting a particular state and action: $r_{s,a}(\mathbf{s}', \mathbf{a}') \triangleq \mathbb{1}(\mathbf{s}' = s, \mathbf{a}' = a)$. This method also fails to learn all vertices of the state marginal polytope, as we show in the following worked example.

We define the dynamics for a 3-state, 2-action MDP, set $\gamma = 0.5$, and define the initial state distribution to be uniform over the three states. This counterexample has deterministic dynamics, represented by arrows below:

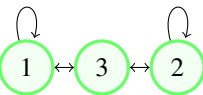

None of the policies that are optimal for reward functions $\{r_{\mathbf{s},\mathbf{a}}\}$ are optimal for the reward $r(\mathbf{s}, a) = \mathbb{1}(\mathbf{s} \in \{1, 2\})$. The intuition behind this counterexample is that there are multiple ways to maximize this reward function: remaining at state 1 and remaining at state 2. However, none of the optimal policies for $\{r_{s,a}\}$ employ this strategy; rather, all these policies deterministically attempt to repeat a single transition.

**Including actions in the mutual information** ($I((\mathbf{s}, a); z)$)**.** Prior work on skill learning has considered many alternative definitions of the mutual information objective, experimenting with conditioning on actions (Eysenbach et al., 2018), initial states (Gregor et al., 2016), and entire trajectories (Co-Reyes et al., 2018). We refer the reader to Achiam et al. (2018) for a survey of these methods. Consider, for instance, the objective $I((\mathbf{s}, \mathbf{a}); \mathbf{z})$. Our reasoning from Lemma 6.3 suggests that this alternative objective would result in learning at most $|\mathcal{S}| \cdot |\mathcal{A}|$ skills, more than the $|\mathcal{S}|$ skills

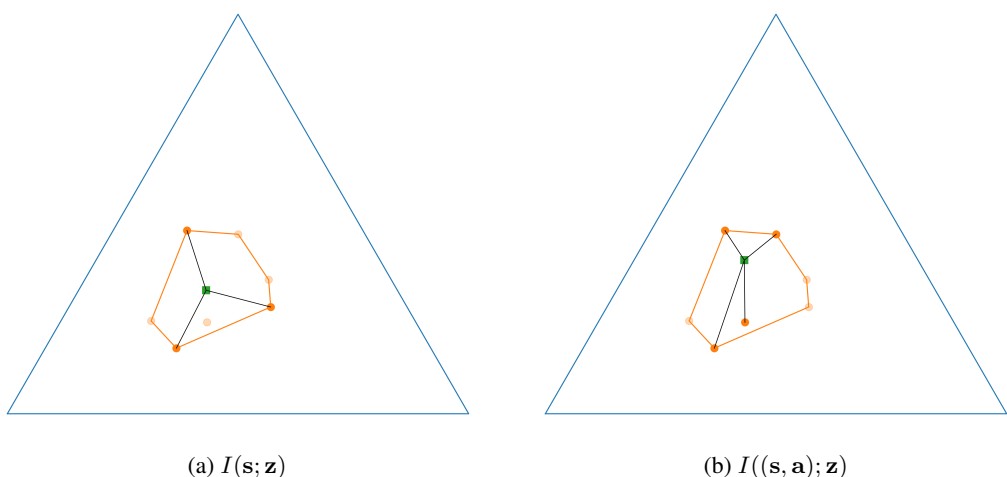

(a) $I(\mathbf{s}; \mathbf{z})$            (b) $I((\mathbf{s}, \mathbf{a}); \mathbf{z})$

Figure 5: **Counterexample 1**: Maximizing $I((\mathbf{s}, \mathbf{a}); \mathbf{z})$ does not result in discovering more vertices of the state marginal polytope.

learned when maximizing $I(\mathbf{s}; \mathbf{z})$. This objective results in discovering vertices of the *state-action* marginal polytope. However, these additional skills do not necessarily cover additional vertices of the *state* marginal polytope. We now present a worked example where maximizing $I((\mathbf{s}, \mathbf{a}); \mathbf{z})$ only covers 3 of 6 vertices, the same number as covered by maximizing $I(\mathbf{s}; \mathbf{z})$.

We describe Counterexample 1 from Sec. 6 in more detail. This example has 3 states, 4 actions, and 7 skills. The matrix describing the state-action marginals $p(\mathbf{s}, \mathbf{a} \mid \mathbf{z})$ is given as follows:

$$P_{sa} = \begin{bmatrix} 0.0716829 & 0.18424628 & 0.1795034 & 0.13672623 & 0.00295358 & 0.09510334 \\ 0.03845856 & 0.05906211 & 0.08669371 & 0.01741351 & 0.02568876 & 0.10246764 \\ 0.11750427 & 0.14607 & 0.11480248 & 0.00287273 & 0.00954288 & 0.10701075 \\ 0.02102705 & 0.02261918 & 0.14991443 & 0.23005368 & 0.06343457 & 0.01514798 \\ 0.13386153 & 0.06171002 & 0.09080011 & 0.16650628 & 0.05625721 & 0.06271223 \\ 0.10353768 & 0.09575361 & 0.02849315 & 0.00634354 & 0.13433545 & 0.05968919 \\ 0.05926209 & 0.10606773 & 0.04541879 & 0.05518233 & 0.14630736 & 0.12098995 \\ 0.05692926 & 0.0745452 & 0.04428352 & 0.12954613 & 0.0261651 & 0.13530253 \\ 0.13704367 & 0.01333569 & 0.10392952 & 0.04119388 & 0.11826303 & 0.12777541 \\ 0.12092377 & 0.0703527 & 0.08563638 & 0.08155578 & 0.04348018 & 0.05650998 \\ 0.06520597 & 0.00354202 & 0.11542116 & 0.09079959 & 0.14099273 & 0.05765374 \\ 0.04477772 & 0.03235234 & 0.07636344 & 0.00824066 & 0.09351822 & 0.27113241 \\ 0.20516233 & 0.19498386 & 0.03909264 & 0.11229717 & 0.06929778 & 0.09957651 \\ 0.02888635 & 0.08654659 & 0.00456412 & 0.10995985 & 0.0328871 & 0.0167457 \end{bmatrix}$$

The matrix describing the state marginals is computed as follows:

$$P_a = P_{sa} \begin{bmatrix} 1. & 0. & 0. \\ 1. & 0. & 0. \\ 1. & 0. & 0. \\ 1. & 0. & 0. \\ 0. & 1. & 0. \\ 0. & 1. & 0. \\ 0. & 1. & 0. \\ 0. & 1. & 0. \\ 0. & 0. & 1. \\ 0. & 0. & 1. \\ 0. & 0. & 1. \\ 0. & 0. & 1. \end{bmatrix}$$

We illustrate the skills found by maximizing $I(\mathbf{s}; \mathbf{z})$ and $I((\mathbf{s}, \mathbf{a}); \mathbf{z})$ in Fig. 5. Note that including the actions in the mutual information does *not* result in discovering more vertices of the state marginal polytope.

## A.6 PROOF OF THEOREM 4.1

*Proof.* Our proof proceeds in two steps. First, we will solve the adaptation problem, assuming that both the reward function $r(\mathbf{s})$ and the initialization $\rho(\mathbf{s})$ are given. We then substitute the optimal policy $\rho^*(\mathbf{s})$ into the pretraining problem and solve for the optimal initialization $\rho(\mathbf{s})$.

First, we solve for the optimal policy $\rho^*(\mathbf{s})$ of the adaptation step. Using calculus of variations, the optimal policy can be written as

$$\rho^*(\mathbf{s}) = \frac{\rho(\mathbf{s})e^{r(\mathbf{s})}}{\int \rho(\mathbf{s}')e^{r(\mathbf{s}')}ds'}.$$

We can then express the cost of adaptation as follows:

ADAPTATIONOBJECTIVE$(\rho(\mathbf{s}), r(\mathbf{s}))$

$$= \max_{\rho^+(\mathbf{s})\in\mathcal{C}} \mathbb{E}_{\rho^+(\mathbf{s})}[r(\mathbf{s})] - \mathbb{E}_{\rho^*(\mathbf{s})}[r(\mathbf{s})] + D_{\mathrm{KL}}(\rho^*(\mathbf{s}) \,\|\, \rho(\mathbf{s}))$$

$$= \max_{\rho^+(\mathbf{s})\in\mathcal{C}} \mathbb{E}_{\rho^+(\mathbf{s})}[r(\mathbf{s})] - \frac{\int r(\mathbf{s})\rho(\mathbf{s})e^{r(\mathbf{s})}ds}{\int \rho(\mathbf{s}')e^{r(\mathbf{s}')}ds'} + \frac{\int \rho(\mathbf{s})e^{r(\mathbf{s})}}{\int \rho(\mathbf{s}')e^{r(\mathbf{s}')}ds'}\left(\log\rho(\mathbf{s}) + r(\mathbf{s}) - \log\int \rho(\mathbf{s}')e^{r(\mathbf{s}')}ds' - \log\rho(\mathbf{s})\right)ds$$

$$= \max_{\rho^+(\mathbf{s})\in\mathcal{C}} \mathbb{E}_{\rho^+(\mathbf{s})}[r(\mathbf{s})] - \frac{\int r(\mathbf{s})\rho(\mathbf{s})e^{r(\mathbf{s})}ds}{\int \rho(\mathbf{s}')e^{r(\mathbf{s}')}ds'} + \frac{\int \rho(\mathbf{s})e^{r(\mathbf{s})}}{\int \rho(\mathbf{s}')e^{r(\mathbf{s}')}ds'}r(\mathbf{s})ds - \log\int \rho(\mathbf{s})e^{r(\mathbf{s})}ds$$

$$= \max_{\rho^+(\mathbf{s})\in\mathcal{C}} \mathbb{E}_{\rho^+(\mathbf{s})}[r(\mathbf{s})] - \log\int \rho(\mathbf{s})e^{r(\mathbf{s})}ds.$$

We can note write the pretraining problem (Eq. 3) as follows:

$$\min_{\rho(\mathbf{s})\in\mathcal{C}} \max_{r(\mathbf{s})\in\mathbb{R}^{|\mathcal{S}|}} \text{ADAPTATIONOBJECTIVE}(\rho(\mathbf{s}), r(\mathbf{s})) = \min_{\rho(\mathbf{s})\in\mathcal{C}} \max_{r(\mathbf{s})\in\mathbb{R}^{|\mathcal{S}|}} \max_{\rho^+(\mathbf{s})\in\mathcal{C}} \mathbb{E}_{\rho^+(\mathbf{s})}[r(\mathbf{s})] - \log\int \rho(\mathbf{s})e^{r(\mathbf{s})}ds.$$

We now aim to solve this optimization problem for $\rho(\mathbf{s})$. We start by noting that the $\max$ operator is commutative, so we can swap the order of the maximization of $r(\mathbf{s})$ and $\rho^+(\mathbf{s})$. That is, instead of the adversary first choosing an reward function and then choosing an optimal policy for that reward function, we can consider the adversary first choosing an optimal policy and then choosing a reward function for which that policy is optimal.

$$\min_{\rho(\mathbf{s})\in\mathcal{C}} \max_{r(\mathbf{s})\in\mathbb{R}^{|\mathcal{S}|}} \text{ADAPTATIONOBJECTIVE}(\rho(\mathbf{s}), r(\mathbf{s})) = \min_{\rho(\mathbf{s})\in\mathcal{C}} \max_{\rho^+(\mathbf{s})\in\mathcal{C}} \max_{r(\mathbf{s})\in\mathbb{R}^{|\mathcal{S}|}} \mathbb{E}_{\rho^+(\mathbf{s})}[r(\mathbf{s})] - \log\int \rho(\mathbf{s})e^{r(\mathbf{s})}ds.$$

Using calculus of variations, we determine that the optimal reward function satisfies

$$r(\mathbf{s}) = \log\rho^+(\mathbf{s}) - \log\rho(\mathbf{s}) + b,$$

where $b \in \mathbb{R}$ is a constant scalar. There are many optimal reward functions, one for each choice of $b$. However, as the scalar $b$ will cancel out in the next step of our proof, this multiplicity is not a concern. Substituting this worst-case reward function, we can write the overall pretraining objective as

$$\min_{\rho(\mathbf{s})\in\mathcal{C}} \max_{r(\mathbf{s})\in\mathbb{R}^{|\mathcal{S}|}} \text{ADAPTATIONOBJECTIVE}(\rho(\mathbf{s}), r(\mathbf{s}))$$

$$= \min_{\rho(\mathbf{s})\in\mathcal{C}} \max_{\rho^+(\mathbf{s})\in\mathcal{C}} \mathbb{E}_{\rho^+(\mathbf{s})}[\log\rho^+(\mathbf{s}) - \log\rho(\mathbf{s}) + b] - \log\int \rho(\mathbf{s})\frac{\rho^+(\mathbf{s})}{\rho(\mathbf{s})}e^b ds$$

$$= \min_{\rho(\mathbf{s})\in\mathcal{C}} \max_{\rho^+(\mathbf{s})\in\mathcal{C}} \mathbb{E}_{\rho^+(\mathbf{s})}[\log\rho^+(\mathbf{s}) - \log\rho(\mathbf{s}) + b] - \log e^b$$

$$= \min_{\rho(\mathbf{s})\in\mathcal{C}} \max_{\rho^+(\mathbf{s})\in\mathcal{C}} \mathbb{E}_{\rho^+(\mathbf{s})}[\log\rho^+(\mathbf{s}) - \log\rho(\mathbf{s})]$$

$$= \min_{\rho(\mathbf{s})\in\mathcal{C}} \max_{\rho^+(\mathbf{s})\in\mathcal{C}} D_{\mathrm{KL}}(\rho^+(\mathbf{s}) \,\|\, \rho(\mathbf{s})).$$

Applying Lemma 6.5 completes the proof. □

## A.7 ASSUMPTION ON CONCYCLIC VERTICES

Lemma 6.3 requires the following regularity condition:

**Assumption 1.** *Assume that any subset of $|\mathcal{S}|$ vertices is not concyclic with respect to the KL divergence. That is, for all subsets of vertices $\mathcal{V} \subseteq \text{VERTICES}(\mathcal{C})$ with size $|\mathcal{S}|$, there does not exist a state distribution $\rho(\mathbf{s})$ that has an equal divergence to all vertices in the subset:*

$$D_{\mathrm{KL}}(\rho_i(\mathbf{s}) \parallel \rho(\mathbf{s})) = D_{\mathrm{KL}}(\rho_j(\mathbf{s}) \parallel \rho(\mathbf{s})) \quad \forall i, j \in \mathcal{V}.$$

This assumption can be understood visually using Fig. 3a *(Left)*. A simplex in $|\mathcal{S}|$ has $|\mathcal{S}| - 1$ degrees of freedom. So, given $|\mathcal{S}| - 1$ points, we can uniquely determine a center that is equidistant from these $|\mathcal{S}| - 1$ points. If we add an additional point to the simplex, it is very unlikely (measure zero) to be the same distance from this center.

### A.8 EXAMPLE WHERE ALL DETERMINISTIC POLICIES ARE UNIQUE VERTICES

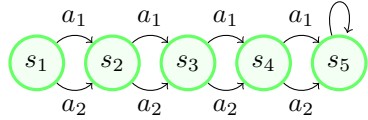

Figure 6

## B EXPERIMENTAL DETAILS AND ADDITIONAL PLOTS

For all figures in the paper, the dynamics were randomly generated. One way to generate the state marginal polytope is to randomly sample the initial state distribution and the dynamics distribution. We employed a simpler, but mathematically equivalent, sampling procedure. We first sampled a set of state marginal distributions, and then computed the convex hull. Note that, given a set of state marginal distributions, it is always possible to construct an MDP whose state marginal polytope is the convex hull of this set. Assign one action to each skill, and define the dynamics as

$$p(\mathbf{s_{t+1}} \mid \mathbf{s_t}, \mathbf{a_t}) = \rho(\mathbf{s} \mid \mathbf{z} = \mathbf{a}) \quad \text{for all states } \mathbf{s_t}.$$

Then, define the initial state distribution to be uniform over all states.

Please see the supplemental Jupyter notebook for code to reproduce the figures. Fig. 7 plots the same experiment as Fig. 3 in the main text, repeated for different randomly-sampled dynamics.

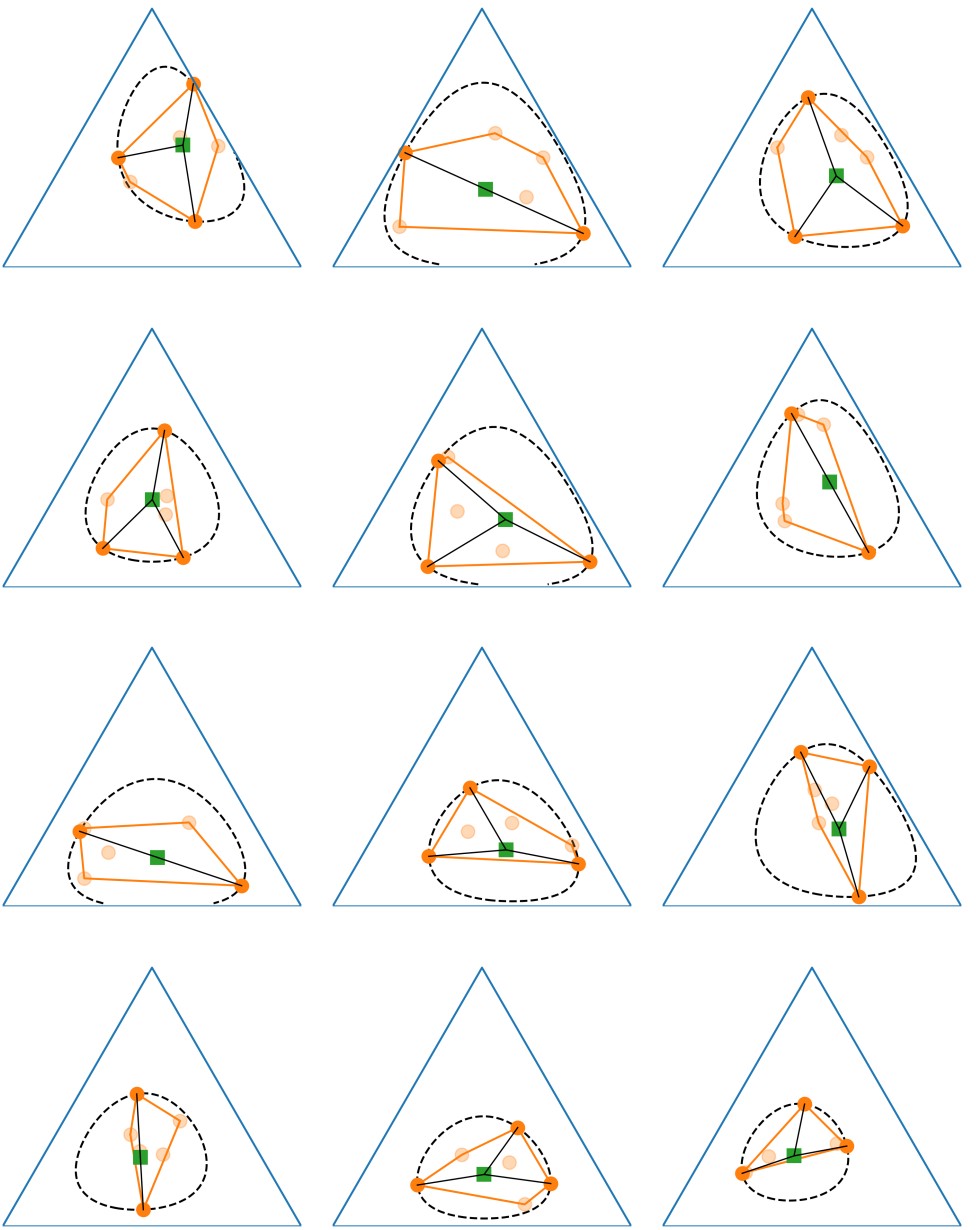

Figure 7: **Information Geometry of Skill Learning**: More examples of the skills learned by MISL.

