# OpenReview forum: "The Information Geometry of Unsupervised Reinforcement Learning"
_ICLR.cc/2022/Conference — ICLR 2022 Oral_

### Official Review · Reviewer_tCVb · 2021-11-02

**Correctness:** 4
**Technical Novelty And Significance:** 4
**Empirical Novelty And Significance:** Not applicable
**Recommendation:** 8
**Confidence:** 4

**Main Review:**

Overall, the paper presents an insightful analysis of some of the challenges of unsupervised RL. In general, I believe that the primary impact of this paper would be through its geometric approach of analyzing the space of learnable policies. The concrete theorems proven by the paper, while interesting, appear to have a relatively large set of conditions. It would be interesting to see how the analysis can be applied to more general settings.

**Summary Of The Paper:**

The paper treats the problem of unsupervised RL, which it defines as the problem of pretraining a system, without having access to a reward function, to learn a collection of policies, that are labeled skills. The idea is that when the reward function is presented, the target policy can be assembled as a combination or composition of these skills. Such policies had been previously proposed to be learned using mutual information maximization approaches.

The primary contribution of this paper that it analyzes the space of policies/skills that are learnable using mutual information based approaches. The authors present this analysis through a geometric lens on the probability simplex of the possible states of the system. A given policy or skill can be associated to a point in this space by associating with a skill or policy its discounted state probability distribution.

Using these analytical tools, the authors prove several interesting results. For instance, they show that by maximizing mutual information alone, one cannot learn sufficient skills to cover all the set of optimal policies - the number of unique skills learned through this approach is bounded by the dimensionality of the state space.

**Summary Of The Review:**

The paper analyzes some of the challenges of unsupervised RL through a geometric approach of analyzing the space of learnable policies. The approach allows insights about the characteristics and certain optimality criteria that applies to skills learned through mutual information type approaches.

---

### Official Review · Reviewer_Xkef · 2021-11-02

**Correctness:** 4
**Technical Novelty And Significance:** 4
**Empirical Novelty And Significance:** Not applicable
**Recommendation:** 8
**Confidence:** 3

**Main Review:**

STRENGTHS

(S1) The paper provides a very nice analysis for a growing class of methods that seek to solve a problem of growing interest in the community. The results are interesting--they say what unsupervised skill discovery methods _can_ and _can not_ do--and the discussion is timely as more and more researchers start to work in the space of unsupervised reinforcement learning.

(S2) The geometric perspective presented is intuitive and does a nice job of communicating the main results of the paper.

WEAKNESSES

(W1) The paper lacks a good empirical confirmation of the claims made here. For example, if unsupervised skill discovery indeed hits the theoretical upper bound that the authors claim, shouldn't there be even a very small toy domain on which the authors could do some experiments to validate that claim? That said, I understand the contribution of the paper here is mostly theoretical in nature.

**Summary Of The Paper:**

The authors provide an in-depth study of unsupervised skill discovery from the RL literature. In particular, the authors analyze such approaches from the perspective of "information geometry," and show that these algorithms _can not_ learn skills that are optimal for all possible reward functions, but _can_ provide a good initialization for online learning approaches that seek to adapt the initial skills to find optimal policies for new reward functions.

**Summary Of The Review:**

Overall, I really like the paper and think that the results are important and should be presented to the community. However, I do think the paper could be improved with even a small empirical study.

---

### Official Review · Reviewer_r6uL · 2021-11-02

**Correctness:** 4
**Technical Novelty And Significance:** 4
**Empirical Novelty And Significance:** Not applicable
**Recommendation:** 8
**Confidence:** 4

**Main Review:**

## Strengths:
* This paper attempts to answer the fundamental question of why RL practitioners should care about unsupervised skill discovery. Providing a satisfying answer to this question should attract more attention to learning skills in this setting, as well as test and clarify the fundamentals on which the problem is based.
* The high level results this paper presents also seem interesting.
* Adapting the value function polytope to a polytope over state visitation frequencies and analyzing the problem from a geometric perspective is also a novel approach that deserves to be studied further.
* The reframing of the unsupervised skill discovery problem as a vertex discovery problem is very interesting.

## Weaknesses:
* I am not completely convinced of the correctness of all the results. The paper introduces a lot of tools for its analysis. Specifically, the state marginal distribution polytope is an entirely new concept and its use in the analysis in this paper was nontrivial to follow. My questions in this regard are below.
* The adaptation procedure that this paper refers to when it asserts that the mutual information maximizing skills are optimal initializations ignores the dynamics of the environment. As such, I am not sure if this adaptation procedure should even be considered feasible. The paper should reframe its statement to make the _strong_ assumption on this adaptation clear, rather than merely stating that the initialization is optimal "under some assumption about how the adaptation is performed".

## Detailed Comments and Questions:
* The problem setup specifies $\gamma \in [0, 1]$. Doesn't $\gamma=1$ not work for the state marginal distribution and the subsequent RL objective?
* The value function polytope is a closely related to the value function polytope approach (Dadashi et al., 2019). This related work seems almost hidden in section 2, and the contrast between the two approaches beyond the fact that one looks at value functions and the other at state marginals is not clarified. I felt like a paragraph clarifying the difference so that a reader familiar with that work might be able to grasp the state visitation polytope better would be useful.
* In the 3-state MDP example considered for this paper, the start state distribution and the dynamics of the environment are not illustrated sufficiently. Does the agent start at all three states with equal probability?
* When analysing this polytope in terms of state-only reward functions, does the origin indicate reward functions that do not prefer any particular state over the other? Are the rewards adjusted to be positive, so that they lie in this quadrant that is being analyzed?
* In general, section 5 seems very condensed. I understand that the limits of the paper might be forcing the authors to be succinct, but I come away from this section feeling like I haven't completely understood the geometric view the paper is utilizing.
* In Section 6, the notion of policies being closer or further in terms of the KL divergence is mentioned when analyzing the skills learned via mutual information maximization. I'm having a hard time wrapping my head around this concept, and perhaps it stems from not grasping the polytope in the previous section completely. Do points further away from each other on the polytope have higher a higher value of KL divergence? But if it was possible for two policies to spend all their time at individual vertices of the outer triangle (if the dynamics of the MDP allowed it), wouldn't the KL divergence be infinite? The notion of treating distance in this polytope in terms of KL divergence is something that I am not convinced of, and leads to me being unconvinced of the rest of the results of this paper.
* In Lemma 6.2, does the probability of sampling a skill ($p(z)$) have to be uniform? I am unsure how skills with different probabilities could lead to state marginals that have the same KL divergence from the average. Do the skills with lower probability of being sampled travel farther?
* For theorem 6.4, if the agent is able to learn $|\mathcal{S}|$ skills that each maximize the time spent at individual states, then one of these skills should maximize the return for any state based reward function. What am I missing here? Why do we need to consider all $|\mathcal{A}|^|\mathcal{S}|$ policies?
* On page 7, after the proof of Lemma 6.3, the paper mentions a regularity assumption. What is this assumption?
* ICLR allows 10 pages of content this year. Given the density of Section 5, I feel like this paper wasted some space that could have been used for some clarifying exposition on the geometric view that the paper proposes.

## Other Comments:
* After Equation (3): "... can be written is the  average state ...", _is_ should be _as_.
* Section 6.3: "Maximizing mutual information pushes skills away from one another, not as finding a skill or policy that is close to every other policy". I'm not sure what this sentence is trying to say. It is hard to parse.

**Summary Of The Paper:**

This paper is trying to analyze whether unsupervised skill discovery is useful for more easily solving any possible downstream tasks in an MDP. It does so by adapting the idea of the value function polytope to a state visitation distribution polytope. It also specifies possible reward functions in this geometric setting and analyzes the connection between points on this polytope and returns with respect to the reward function.

Next, it casts the mutual information based skill discovery problem in this geometric space and tries to analyze the skills learned. From their analysis, the paper suggests ways to infer how many skills can be learned, and whether those skills are optimal with respect to some downstream tasks.

It seems that these skills can be guaranteed to be the vertex of the above polytope, but not to be optimal with respect to all downstream reward functions. They then suggest that the skills learned might be useful for an adaptation procedure that ignores the dynamics of the environment.

**Summary Of The Review:**

The problems that this paper attempts to solve are interesting and would add value to the setting of unsupervised skill discovery. Specifically, it considers the question of what the objective of such skill discovery should be.
However, the geometric tools introduced in the paper and the subsequent analysis does not seem polished enough and is hard to follow. I am unconvinced of the analysis and thus scoring the paper conservatively.

---

### Decision · Program_Chairs · 2022-01-20

**Decision:**

Accept (Oral)

**Comment:**

Strong submission that analyses the unsupervised skill discovery setting from the perspective of information geometry, which leads to some interesting conclusions. In particular, it is shown that this does not lead to skills that are optimal for all reward functions, but does provide a good initialization for methods that aim to find optimal policies.

Across the board, the reviewers believe the analysis provided by this work is both important and novel. And while there were some initial concerns raised, such as lack of empirical confirmation of some of the claims and some questions about the analysis, the authors have addressed all of these concerns convincingly.

Hence, I strongly recommend acceptance of this submission.